# GoalLadder: Incremental Goal Discovery with Vision-Language Models

**Alexey Zakharov**
University of Oxford
alexey.zakharov@cs.ox.ac.uk

**Shimon Whiteson**
University of Oxford
shimon.whiteson@cs.ox.ac.uk

## Abstract

Natural language can offer a concise and human-interpretable means of specifying reinforcement learning (RL) tasks. The ability to extract rewards from a language instruction can enable the development of robotic systems that can learn from human guidance; however, it remains a challenging problem, especially in visual environments. Existing approaches that employ large, pretrained language models either rely on non-visual environment representations, require prohibitively large amounts of feedback, or generate noisy, ill-shaped reward functions. In this paper, we propose a novel method, **GoalLadder**, that leverages vision-language models (VLMs) to train RL agents *from a single language instruction in visual environments*. GoalLadder works by incrementally discovering states that bring the agent closer to completing a task specified in natural language. To do so, it queries a VLM to identify states that represent an improvement in agent's task progress and to rank them using pairwise comparisons. Unlike prior work, GoalLadder does not trust VLM's feedback completely; instead, it uses it to rank potential goal states using an ELO-based rating system, thus reducing the detrimental effects of noisy VLM feedback. Over the course of training, the agent is tasked with minimising the distance to the top-ranked goal in a learned embedding space, which is trained on unlabelled visual data. This key feature allows us to bypass the need for abundant and accurate feedback typically required to train a well-shaped reward function. We demonstrate that GoalLadder outperforms existing related methods on classic control and robotic manipulation environments with the average final success rate of $\sim$95% compared to only $\sim$45% of the best competitor.

## 1 Introduction

Reinforcement learning (RL) relies critically on effective reward functions to guide agents toward desired behaviours. However, crafting reward functions by hand requires significant human labour and domain expertise. Fortunately, many RL tasks can be succinctly described in natural language (e.g., 'open the drawer', 'close the window'), from which humans can understand the task at hand and approximate a reward function to reach the goal state. In particular, extracting a reward function from a language instruction requires understanding the semantics of the task description, associating its components with the given environment, and integrating this information with prior, common-sense knowledge. The rise of large language models (LLMs) brings hope of automating this process – given a language instruction, can we automatically extract an effective reward function?

Indeed, LLMs can perform well on open-ended question answering, code generation, and be vast reservoirs of common-sense knowledge [1, 2, 3, 4]. Moreover, multimodal models have even broader applicability to tasks involving images or other modalities [5, 6, 7, 8, 9, 10]. These properties make pretrained language models potentially suitable for providing learning signals to reinforcement learning algorithms, such as by generating preference-based feedback [11] or outputting a reward function directly [12].

39th Conference on Neural Information Processing Systems (NeurIPS 2025).

Prior work attempts to use LLMs to generate reward functions given access to environment code or interpretable state representations [13, 14, 12], limiting their applicability. Another line of work uses vision-language models (VLMs) to define or learn a reward function given a language instruction and visual observations. In particular, two broad strategies have emerged: (i) embedding-based [2, 15, 16, 17], which aligns visual observations with a language instruction in the embedding space of a VLM; and (ii) preference-based [11], which prompts a VLM to rank segments of an agent's behaviour by how well they match a textual specification, and trains a reward function from the collected preference data.

Both approaches present certain challenges. Embedding-based methods employing CLIP [5] tend to yield noisy reward functions due to the mismatch between the CLIP training data and the observations from the tested environments [16]. Similarly, these methods require that every observation is embedded to produce rewards, making them expensive and inefficient in using large pretrained models. Preference-based methods like RL-VLM-F [11] produce reward functions with less noise and better correlation with task progress but are nevertheless substantially noisy due to mistakes made by a VLM when comparing trajectories. Erroneous feedback plagues the preference label dataset (at an unknown frequency) making it challenging to robustly train a well-shaped reward function without overfitting to mislabelled samples. Furthermore, although preference-based approaches are more sample-efficient than embedding-based ones, they nevertheless require many VLM queries to train a reward function that generalises well.

In this paper, we argue that, to be effective in practice, approaches employing VLMs for feedback must address two key issues: (a) robustness to noisy VLM feedback; and (b) query efficiency in using VLMs for feedback, given that repeated large-scale queries to VLMs can be prohibitively expensive.

To this end, we propose **GoalLadder**, an algorithm designed to address both of these challenges. Our method leverages a VLM to incrementally discover environment states that bring the agent closer to completing a task. To avoid the problems of prior methods, GoalLadder performs repeated comparisons of a small subset of visual observations and maintains a persistent, ELO-based utility measure (*rating*) over states. This process allows GoalLadder to progressively refine its estimates of state utilities without being severely affected by noisy VLM feedback. Furthermore, compared to previous methods, GoalLadder requires substantially fewer VLM queries to learn desired behaviours, since rewards are defined as distances between visual observations in a learned embedding space that is trained on unlabelled data, allowing for reward generalisation to unseen states without explicit feedback. Using two classic control and five robotic manipulation tasks, we demonstrate that our method significantly outperforms prior work that utilises vision-language models for RL, with the average final success rate of ∼95% compared to only ∼45% of the best competitor. GoalLadder exhibits impressive performance even against the oracle agent that has access to ground-truth reward – nearly matching it across all tested tasks and convincingly surpassing it on one.

## 2   Related Work

**Vision-language models**   Recent advances in large language models [18, 19, 20] demonstrate impressive abilities for reasoning and common sense in text and other modalities such as vision [5, 6, 7, 8, 9, 10]. Vision-language models are trained on a joint visual and textual representation space, allowing the capabilities of LLMs to be applied within the visual domain. Nevertheless, existing VLMs suffer from poor spatial understanding [21], making it difficult to reliably apply them to spatial reasoning tasks. Our approach takes this into account by design, using a robust pairwise evaluation strategy to reduce the effects of erroneous feedback on spatial tasks.

**Language models in RL**   The use of language models in reinforcement learning has been predominantly about making LLMs write code, particularly to design a reward function [13, 14, 12]. Although these approaches may improve as language models get better at code comprehension, it is unclear how they would perform in arbitrarily complex physical simulations or transfer to real-world environments. LLMs have also been used to provide a reward signal in text-based environments [22, 23].

**Vision-language models in RL**   Several works explore the integration of VLMs in RL. One avenue is using the embeddings of the VLMs directly in the definition of the reward. For example, [16] use the CLIP model [5] to embed the task description specified in text, as well as an environment's visual

observations, and define the reward as the cosine similarity between the image and text embedding. However, the resulting reward functions tend to be noisy, which is often attributed to the mismatch between the training data of the VLMs and the environments they are applied to [16]. To this end, Baumli et al. [17] fine-tune a CLIP model with contrastive learning to output the success of an agent in reaching a text-based goal. Other papers use vision-language models to *align* visual observations with language descriptions of a task [2, 24, 15, 25], to perform incremental trajectory improvement from human feedback [26] or to elicit better task-relevant representations [27].

VLMs can also be used for direct feedback to learn a reward function from VLM goal-conditioned preferences [11]. In particular, Wang et al. [11] use VLMs to provide preference labels over states of visual observations conditioned on a task description and use these labels to learn a reward function with Reinforcement Learning from Human Preferences (RLHF) [28]. Although this work improves upon previous methods, noise in the learned reward function remains evident. Furthermore, for the experiments with robotic manipulation, the authors remove the robot from the image to improve the VLM's ability to identify goal states in object-oriented tasks. This trick, however, also reduces the amount of information a VLM can use to identify incrementally better states. In contrast, GoalLadder solves the tasks without environment modifications and utilises an ELO-based rating system to reduce the effects of noisy VLM feedback.

## 3 GoalLadder

### 3.1 Preliminaries

**Reinforcement learning**   We formulate the problem as a Markov Decision Process (MDP), defined as a tuple $\mathcal{M} = \{\mathcal{S}, \mathcal{A}, \mathcal{R}, \mathcal{P}, \gamma\}$. Here, $\mathcal{S}$ represents the state space of the environment, $\mathcal{A}$ is the agent's action space, $\mathcal{P} : \mathcal{S} \times \mathcal{A} \times \mathcal{S} \rightarrow [0, 1]$ defines the transition dynamics, $\mathcal{R} : \mathcal{S} \times \mathcal{A} \rightarrow [0, \infty)$ is the reward function, and $\gamma \in [0, 1)$ is the discount factor. At each timestep $t$, an agent takes an action $a_t$, transitions to state $s_{t+1}$, and receives reward $r_t$. In our setup, we assume that the agent similarly receives an image observation $o_t$ that represents the state of the environment. The goal of the agent is to maximise the sum of discounted rewards in the environment, defined as $G = \sum_{k=0}^{\infty} \gamma^k r(s_k, a_k)$.

**Soft-Actor Critic**   We use soft-actor critic (SAC) [29] as the RL backbone of our agent. However, in GoalLadder, the reward function changes periodically as new targets are being set over the course of training. The vanilla implementation of SAC can struggle with non-stationary reward given that it reuses past data from the replay buffer, which may not be up to date with respect to the latest reward function. As such, similar to [30], we periodically relabel all of the agent's trajectories with the latest reward function.

**Assumptions**   We assume access to a language instruction, $l$, that succinctly describes the task of the agent in any given environment (e.g., "open the drawer"). Furthermore, we consider tasks whose goal can be represented by a visual observation (not necessarily unique).

### 3.2 Method

**Overview**   We propose GoalLadder, which uses a vision-language model to incrementally discover states that bring the agent closer to completing a task specified in natural language. GoalLadder uses a VLM in two ways: (i) to identify *candidate* goals, denoted as $g$, to be put into a ranking buffer $\mathcal{B}_g = \{g_1, g_2, ...\}$ for further evaluation, and (ii) to rate the candidate goals, with rating denoted as $e$, based on their proximity to the goal state specified in language. The candidate goals are discovered and ranked *online*, as the RL agent trains and collects new observations according to its latest SAC policy. Over the course of training, the top-ranked goal serves as the target state that the RL agent is tasked with achieving.

GoalLadder involves several stages that are expanded upon in Sections 3.2.1-3.2.3:

1. *Collection*: the RL agent collects an episode by interacting with the environment according to its current SAC policy.

2. *Discovery*: a VLM is queried to determine whether any of the collected observations represent an improvement over the current top-rated candidate goal.

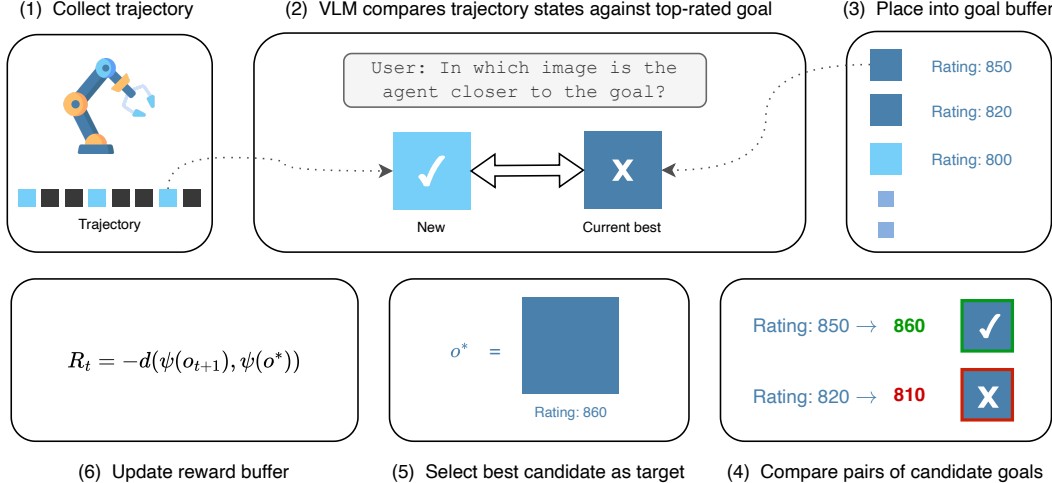

Figure 1: **Summary of GoalLadder. (1-2)** An agent collects a trajectory using the current SAC policy. Trajectory observations are uniformly sampled and compared against the current best candidate goal in the buffer. **(3)** If successful, they are placed into the candidate goal buffer. **(4)** Pairs of candidate goals are sampled and compared against each other using a VLM. Their ratings are updated after each comparison. **(5-6)** Periodically, the target state is updated with the current top-rated goal from the candidate buffer. Reward function is defined as the negative distance to the current best candidate goal in a learned embedding space; the RL agent is then trained with this reward. The process returns to step 1 and repeats.

3. *Ranking*: a VLM is queried with sampled pairs of existing candidate goals from the buffer for comparison and their ratings are updated accordingly.

4. *Training*: The agent is trained to minimise the distance to the top-ranked candidate goal in a learned embedding space.

Together, the stages represent a repeating process, in which the RL agent trains and collects new observations, while candidate goals are being discovered and ranked. A visual illustration of GoalLadder operations is provided in Figure 1.

### 3.2.1 Discovery of candidate goals

To succeed, GoalLadder must address the question: *Which states are worth evaluating as candidate goal states?* Since our capacity to query the VLM is limited, we need to filter out irrelevant states before they are entered into the goal buffer for detailed evaluation. This ensures that the VLM focuses on the most promising states, avoids wasting compute on obvious or trivial cases, and more quickly converges on the final goal. To this end, we maintain a buffer of candidate goals, $\mathcal{B}_g$, where each candidate goal $g_i$ consists of an image $o_i$ representing the state and the goal's rating $e_i$, such that $g_i = (o_i, e_i)$. Given the current top-rated candidate goal,

$$g^* = \arg \max_{g_i \in \mathcal{B}_g} e_i, \quad \text{where } g^* = (o^*, e^*),$$

and a randomly sampled state image $o_j$ from the agent's latest collected trajectory, the VLM is asked to decide which of the two images, $o^*$ or $o_j$, represents a state that is closer to the language-specified task goal, $l$. We denote the VLM output by

$$y = \text{VLM}(o^*, o_j, l), \quad y \in \{-1, 0, 1\},$$

where $y = -1$ indicates that there is no decision or the images are equivalent, $y = 0$ means $g^*$ is deemed a better goal than $o_j$, and $y = 1$ means $o_j$ is deemed a better goal than $g^*$.

If $y = 1$, the newly sampled state $o_j$ is inserted into $\mathcal{B}_g$ as a fresh candidate goal with a standard initial rating. Conversely, if $y = 0$ or $y = -1$, $o_j$ is disregarded in order to keep the goal buffer focused on higher-quality candidates.

### 3.2.2 Rating of candidate goals

Once the goal buffer is large enough, GoalLadder begins querying a VLM to compare the candidate goal states and get a better estimate of their true rating. Specifically, a pair of candidate goals is sampled, $(g_i, g_j) \sim \mathcal{B}_g$, and the same querying process, as in Section 3.2.1, is used.

Ideally, our goal ratings would have two main properties: (1) be robust to noisy VLM outputs and (2) allow for quick updates to a candidate goal's rating when new and better goals are discovered. In GoalLadder, we draw inspiration from the ELO chess rating system [31] to maintain a robust, adaptive measure of each candidate goal's utility. In chess, players' ratings are updated after every match based on two factors: the observed outcome (win, lose, or draw) and the expected outcome given the players' current ratings.

In ELO, the expected score for candidate goal $g_i$ against $g_j$ is given by the logistic function:

$$E_i = \frac{1}{1 + 10^{(e_j - e_i)/C}},$$

where $e_i$ and $e_j$ are the current ratings of $g_i$ and $g_j$, respectively, and $C$ is a constant controlling how sensitive the expected scores are to rating differences.

Given an observed result $S_i \in \{-1, 1, 0\}$ ("win," "loss," or "draw"), we update the rating of $g_i$ as:

$$e_i \leftarrow e_i + T(S_i - E_i),$$

and similarly for $g_j$. Here, $T$ is a constant that governs how quickly ratings are adjusted.

The ELO rating system incrementally absorbs noisy VLM comparisons, adaptively adjusts ratings when new evidence shows a goal is better or worse, and continuously refines its estimates to converge on a stable hierarchy of candidate goals aligned with the language-specified target.

### 3.2.3 Defining the reward function

Once a candidate goal $g^* \in \mathcal{B}_g$ emerges as the highest-rated goal, we treat its image $o^*$ as the agent's current best guess for the true task objective. As such, we wish to encourage the agent to bring its state as close as possible to $o^*$ in an appropriate metric space. However, defining a distance directly in an environment's intrinsic state-space may be unreliable given environment-to-environment differences.

To address this, GoalLadder learns a visual feature extractor to produce a compact latent representation of environment observations. We achieve this using a variational autoencoder training objective [32],

$$\mathcal{L} = -\mathbb{E}_{\psi(z_t|o_t)} \log p_\theta(o_t \mid z_t) + D_{\mathrm{KL}}\left(\psi(z_t \mid o_t) \,\|\, p(z_t)\right), \tag{1}$$

where the encoder $\psi(\cdot)$ is trained to produce a latent vector $z_t$ representing observation $o_t$. We implement the feature extractor using a simple convolutional neural network architecture, the details of which can be found in Appendix A.

By training on a diverse set of observations the agent encounters, we obtain a latent space that captures salient visual (and potentially semantic) features of the environment. Finally, we define the agent's reward at time step $t - 1$ to be:

$$R(s_{t-1}, a_{t-1}) = -d(z_t, z^*),$$

where $z_t = \psi(o_t)$, $z^* = \psi(o^*)$, and $d(\cdot, \cdot)$ is the Euclidean distance. As such, the agent is incentivised to minimise the distance to the top-rated goal $g^*$.

As the candidate goals are being discovered and rated during training, we periodically update our reward function and relabel all stored transitions [30] with respect to the top-rated candidate goal. Periodicity ensures that sudden or erroneous changes in the top-rated goal do not significantly destabilise the training process, while also keeping the best goal estimate up-to-date.

### 3.2.4 Implementation details

We use Gemini 2.0 Flash[1] as our VLM backbone. The top-rated candidate goal is selected as the new target every $L = 5000$ environment steps. The goal buffer size is capped at $|\mathcal{B}_g| = 10$, as the lowest

---

[1]`https://ai.google.dev/gemini-api/docs/models#gemini-2.0-flash`

**Algorithm 1:** GOALLADDER: Pseudo algorithm

---

**Input** : Task description $l$, experience buffer $\mathcal{D}$, SAC, VLM, reward update schedule $L$

1 Initialise candidate goal buffer $\mathcal{B}_g$ with random observations
2 **while** *training* **do**
3     Collect episode $\mathcal{E} \sim \pi$ and store in $\mathcal{D}$
4     `// Discovery and Ranking`
5     **if** $t \bmod K = 0$ **then**
6        Sample $M$ trajectory observations $\{o_1, \ldots, o_M\} \sim \mathcal{E}$
7        Compare each $o_i \in \{o_1, \ldots, o_M\}$ against top-rated candidate using VLM and $l$
8        Update goal buffer $\mathcal{B}_g$ based on the feedback
9        Compare $M$ pairs of $(g_i, g_j) \sim \mathcal{B}_g$ using VLM and $l$
10        Update candidate goal ratings, $(e_i, e_j)$
11     **end**
12     `// Update reward function`
13     **if** $t \bmod L = 0$ **then**
14        Select the top-rated goal from the buffer, $g^* = \arg\max_{(o_i, e_i) \in \mathcal{B}_g} e_i$
15        Update rewards in experience buffer $\mathcal{D}$ using $\|\psi(o) - \psi(o^*)\|_2$
16     **end**
17     `// Train`
18     **if** $t \bmod 1 = 0$ **then**
19        Update SAC
20        Update $\psi$ using a minibatch of observations from $\mathcal{D}$
21     **end**
22 **end**

---

rated candidate goals are removed every $L$ steps. This ensures the comparisons remain focused on the most promising states. VLM queries are performed every $K$ environment steps with $M$ queries per feedback session (see Section 4). SAC gradient update is performed after every environment step. Finally, the VLM prompts are standardised using a single template to ensure consistency and ease of use . Algorithm 1 shows the stepwise, high-level operations of GoalLadder. See Appendix A for further implementation details, including hyperparameters, architectural details, and used computational resources.

## 4 Experiments

We analyse the performance of our algorithm in the scope of continuous control tasks, from classic control to more complex robotic manipulation. Specifically, we aim to answer the following questions:

1. Can GoalLadder effectively *discover* the best goal over the course of training?
2. Can GoalLadder *solve* continuous-control tasks?
3. Is GoalLadder more *sample-efficient* with respect to VLM queries than related methods?

**Environments** We run a variety of continuous-control experiments to investigate GoalLadder's performance. In particular, we use two classic control environments (*CartPole, MountainCar*) from OpenAI Gym [33] and five robotic manipulation environments (*Drawer Close, Drawer Open, Sweep Into, Window Open, Button Press*) from the Metaworld suite [34]. Importantly, unlike prior works [16, 11], we do not perform any special environment modifications to help VLMs discern task progress. We use feedback rates of $K = 2000$ with $M = 5$ for OpenAI Gym environments and $K = 500$ with $M = 5$ for the Metaworld environments.

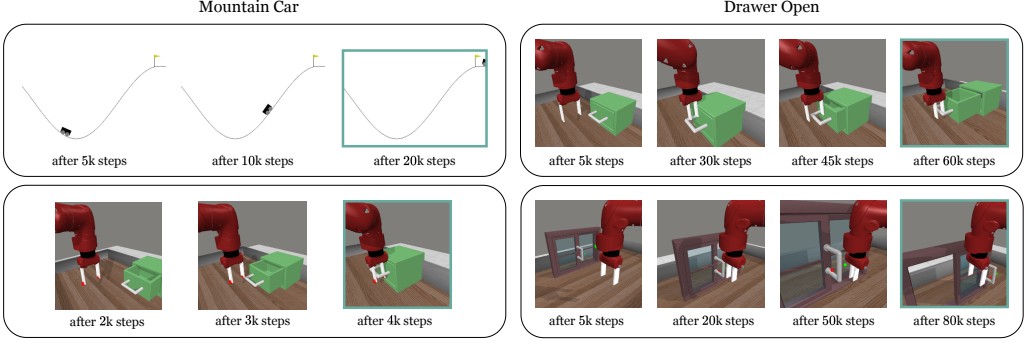

Figure 2: **Top-rated candidate goals during training.** Top-rated candidate goals in the Mountain Car environment follow a natural progression of the car getting closer to the top of the hill. Similarly, in the Metaworld tasks, GoalLadder rapidly discovers states that bring the robotic arm closer to the true task objective, until the best goal is discovered.

**Baselines**    We compare GoalLadder to the following baselines:

- **Ground-truth reward (Oracle)**: A strong baseline in which the RL algorithm is trained with the environment's intrinsic reward. This benchmark acts as the oracle and should serve as an upper bound on the performance of our method.

- **VLM-RM** [16]: Uses CLIP [5] to embed the task description and an image observation and defines the reward as the cosine-similarity between the two embeddings.

- **RoboCLIP** [15]: Employs a pre-trained video-language model, S3D [35], to compute rewards based on the similarity between an agent's video trajectory and either a reference video or a text description. To ensure a fair comparison, we use the text version of RoboCLIP.

- **RL-VLM-F** [11]. Uses a VLM to compare pairs of images with respect to a language instruction. Unlike GoalLadder, RL-VLM-F utilises VLM feedback to train a reward function from the collected labels. For this baseline, we apply *the same feedback rate* used in GoalLadder and use the same vision-language model as backbone – Gemini 2.0 Flash.

## 4.1    Can GoalLadder discover the best goal?

Figure 2 shows the evolution of the top-rated candidate goals in the buffer over the course of the training for the *MountainCar* and *Metaworld* environments. Despite the VLM's tendency to make mistakes in its comparisons, the best goal consistently rises to the top of the ranking. Based on our experiments, we confirm that GoalLadder can effectively discover the target goal given the language instruction for all tested environments.

Similarly, Figure 3 shows the evolution of the candidate goal ratings in the buffer for the *Metaworld Window Open* environment. During the initial discovery stage (until 50k steps), there is no clear winner and the ratings tend to be similar. Upon the discovery of a clear winner (at around 50k steps), GoalLadder quickly singles it out as the best goal for the agent to pursue. Figure 5 provides further insight by visualising the entire goal buffer over the course of training for the *Drawer Open* task. Two trends can be observed: (i) the buffer is approximately ordered by how close the agent is to achieving the task in each image, and

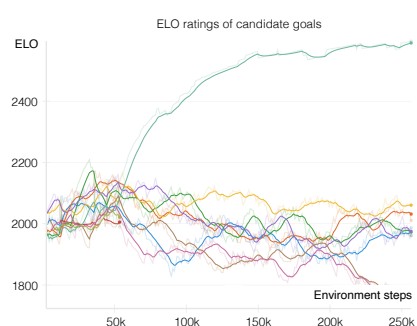

Figure 3: **Candidate goal ratings over time in *Window Open*.** Different colours indicate different candidate goals present in the buffer. GoalLadder quickly singles out the best goal once it is discovered.

(ii) the buffer becomes progressively filled with better states as training goes on. We observe similar behaviour for all other environments.

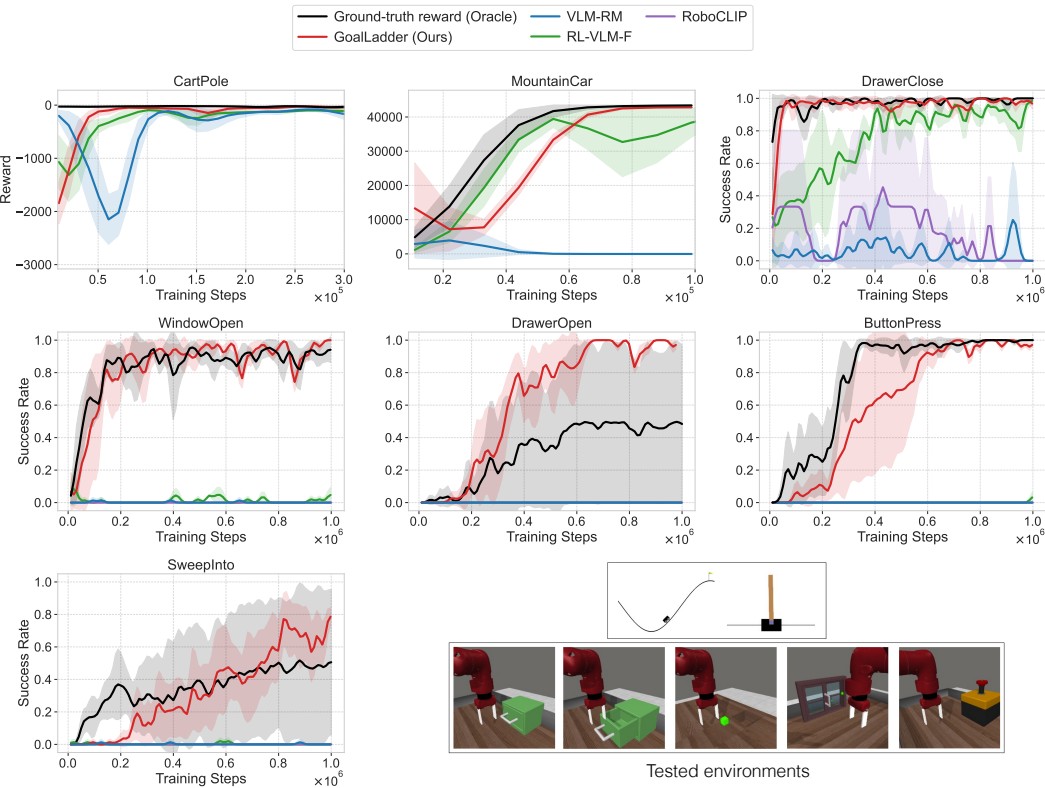

Figure 4: GoalLadder performance against baselines on OpenAI Gym and Metaworld environments. Shaded regions represent the standard deviation over 3 seeds. Averaged across all tasks, GoalLadder achieves a mean success rate of ∼95%, compared to the next best competitor performance of ∼45% achieved by RL-VLM-F.

## 4.2 Can GoalLadder solve the tasks?

Figure 4 shows that GoalLadder can effectively solve continuous control tasks given only a language instruction, reaching a mean success rate of ∼95% averaged across all tasks. Curiously, our method nearly matches the performance of the ground-truth reward baseline and even convincingly surpasses it for the *Drawer Open* task. Empirically, we find that sometimes the oracle agent learns to reach the drawer, but does not learn to pull the handle to open it. We believe this once again highlights the difficulty of manually designing effective reward functions – an issue that was also previously discussed in Wang et al. [11].

The preference-based baseline, RL-VLM-F, manages to solve the relatively easy tasks – *Cartpole* and *MountainCar* from the OpenAI Gym suite, as well as *Drawer Close* from Metaworld. However, it begins to struggle with the more complicated Metaworld tasks. We attribute this poor performance of RL-VLM-F to the fact that it requires more feedback to learn an effective reward function, as well as ways to mitigate the effects of noisy labels in the preference data. Similarly, VLM-RM and RoboCLIP struggle to learn in most of the tasks, which is in line with previously observed limitations of these methods [11].

## 4.3 How sample-efficient is GoalLadder?

Using large pretrained models is expensive, so methods that rely on them for feedback must do so efficiently. While embedding-based approaches, like VLM-RM [16], use all environment observations for reward calculation, preference-based methods, like RL-VLM-F [11], can offer better sample efficiency by training a reward function from a limited amount of feedback, in the hope that the learned reward function generalises to unseen states. By contrast, GoalLadder produces reward signals that generalise to unseen states by defining the reward in terms of distances in an embedding space (Section 3.2.3) trained with unsupervised learning. Utilising the *full* extent of agent's experiences, rather than just the labelled observations, for reward definition allows GoalLadder to approximate state utility without collecting large amounts of feedback. Averaged across all *Metaworld* tasks, GoalLadder

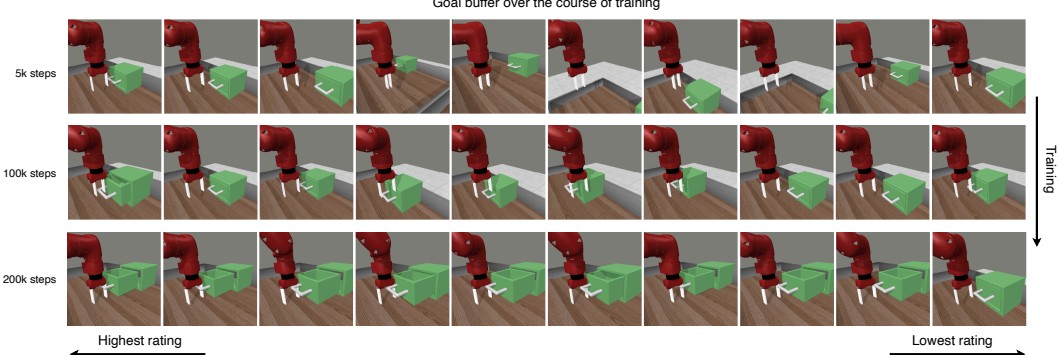

Figure 5: Visualisation of the goal buffer in GoalLadder over the course of training. Each line demonstrates the candidate goals present in the buffer after an indicated number of environment steps. The candidate goals are ordered by ELO rating, from left to right.

learns to solve them after only ∼4500 queries. For comparison, PEBBLE [30], a preference-based method that uses *ground-truth reward* preferences, achieves the same performance after attaining an average of ∼15000 preference labels on the same tasks. Indeed, we observe that GoalLadder boasts superior sample efficiency compared to the tested baselines. Performance curves in Figure 4 illustrate that RL-VLM-F struggles to learn the tasks using the feedback rate of GoalLadder, while VLM-RM, that embeds all collected observations, fails almost entirely.

## 5 Discussion

### 5.1 Limitations

As mentioned in Section 3.2, we assume that the progress or success of a task can be identified from a single image, limiting the application of our method to environments with static goals. Nevertheless, we believe that future work could extend GoalLadder to video-based settings. GoalLadder also largely relies on visual feature similarity for reward definition. In some settings, visual similarity between observations could be a limiting proxy for the underlying state similarity or task progress. Here, more advanced visual embedding techniques could be used instead. Lastly, while our method could always benefit from evaluation on more environments, the costs of using VLMs limit what is feasible.

### 5.2 Broader impacts

Reinforcement learning from language instructions opens up the door for human-interpretable ways to interact with robotic systems in the real world. At the same time, robotic systems trained with the assistance of large pretrained language models have the potential to inherit any existing biases of these models. Though the scope of this paper is limited to object-oriented environments, we would caution against the application of such models in settings with real humans, until this potential issue is thoroughly investigated.

### 5.3 Conclusion

We introduced GoalLadder, a method for training reinforcement learning agents from vision-language model feedback using a single language instruction. GoalLadder systematically discovers new and better states that take the agent closer to the final task goal over the course of training. Our method demonstrated impressive performance gains against prior work, while nearly matching the performance of the oracle agent with access to ground-truth reward. We argued that implementing a more comprehensive pairwise comparison technique and using a learned embedding space for reward definition allowed GoalLadder to be robust to noisy feedback and significantly more sample-efficient compared to prior methods.

GoalLadder offers several interesting directions of future research. Firstly, we believe that our method could be extended to video-based settings, depending on the capabilities of the existing vision-language models for video comprehension. Secondly, we believe GoalLadder would benefit from more advanced visual feature extraction techniques or from building a more meaningful latent space, for example using self-supervised learning.

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

# A Implementation details

## A.1 Soft-Actor Critic

The implementation of the Soft-Actor Critic (SAC) [29] used in the paper follows the implementation of SAC in PEBBLE [30], similar to Wang et al. [11] to ensure a fair comparison. As such, the used hyperparameters, including the used optimisers, are the same and can be found in the original paper [30].

## A.2 Feature extractor

As mentioned in the paper, we train a visual feature extractor $\psi(\cdot)$ using a variational autoencoder (VAE) training objective. In particular, the VAE is implemented using a simple convolutional neural network architecture, the details of which are shown in Table 1. The dimensionality of latent states $|z|$ was chosen to be 16, thus ensuring high informational bottleneck to encourage feature disentanglement. The feature extractor is trained using Adam optimiser [36] with a learning rate of 0.0001. The batch size is 128. A gradient step is applied after each environment step, and the training continues throughout RL training. We set the beta parameter $\beta = 0.1$, which is a common way to encourage better image reconstructions in VAEs [37], and use MSE as reconstruction loss.

To ensure training stability with respect to the RL target (top-rated candidate goal), we fix the weights of the VAE for calculating visual embeddings and update them every $L = 5000$ steps – at the same rate as top-rated candidate goals are updated as RL target states.

Table 1: Architecture of the convolutional feature extractor.

| Component | Layer | Output Shape |
|---|---|---|
| Encoder | Conv2D(3, 16, 4x4, stride=2, pad=1) + ReLU
Conv2D(16, 32, 4x4, stride=2, pad=1) + ReLU
Conv2D(32, 64, 4x4, stride=2, pad=1) + ReLU
Conv2D(64, 128, 4x4, stride=2, pad=1) + ReLU
Conv2D(128, 256, 4x4, stride=2, pad=1) + ReLU
Conv2D(256, 256, 4x4, stride=2, pad=1) + ReLU
Flatten | 128x128x16
64x64x32
32x32x64
16x16x128
8x8x256
4x4x256
4096 |
| Encoder (FC) | Linear(4096 → 16) | 16 |
| Decoder (FC) | Linear(16 → 4096) | 4096 |
| Decoder | ConvT2D(256, 256, 4x4, stride=2, pad=1) + ReLU
ConvT2D(256, 128, 4x4, stride=2, pad=1) + ReLU
ConvT2D(128, 64, 4x4, stride=2, pad=1) + ReLU
ConvT2D(64, 32, 4x4, stride=2, pad=1) + ReLU
ConvT2D(32, 16, 4x4, stride=2, pad=1) + ReLU
ConvT2D(16, 3, 4x4, stride=2, pad=1) + Sigmoid | 8x8x256
16x16x128
32x32x64
64x64x32
128x128x16
256x256x3 |

## A.3 GoalLadder

### A.3.1 Candidate goal rating

When a candidate goal gets added to a goal buffer $\mathcal{B}_g$, it gets assigned an initial rating, $\hat{e}$. The initial rating is calculated by taking the mean rating of all existing goals in the buffer, $\hat{e} = \sum_{e_i \in \mathcal{B}_g}^{N} e_i / N$. This ensures that new goals are able to quickly catch up with top-rated goals, while not being immediately considered superior upon initial discovery.

We also set the parameters of ELO rating updates: $C = 400$ (constant controlling how sensitive the expected scores are to rating differences) and $T = 32$ (rating update speed). These were chosen to match the commonly used values in chess rating. In practice, we find that these default parameters were satisfactory for effectively absorbing noisy VLM outputs, while bringing the best goal to the top of the buffer.

### A.3.2 Reward definition

Rewards are calculated using Euclidean distance between visual embeddings, $\|\psi(o) - \psi(o^*)\|_2$. In practice, we find that it is useful to normalise rewards within bounds $[0, 1]$, using a simple max-min normalisation, and then apply a non-linear transformation, such that the rewards used during training, $\hat{r}$ are calculated as $\hat{r} = r^\gamma$, where $\gamma = 20$. This ensures that reward scaling does not significantly impact the learning process of the RL agent, as the visual feature extractor changes over the course of training. The non-linear transformation ensures that the agent accrues proportionally larger amounts of reward the closer it gets to the target state.

### A.4 Computational resources

For training GoalLadder, we use Tesla V100 16GB GPU and 2× Intel Xeon E5-2698 v4 CPUs, each with 20 cores and 2 hardware threads per core. The training time of a single GoalLadder agent took ∼45 hours. Initial experimentation included identifying the abilities of the used vision-language model, Gemini 2.0 Flash, to compare pairs of visual observations with respect to a language instruction.

## B Task descriptions and prompts

Below is the prompt template used to get feedback from a vision-language model. The formatting instructions of the expected response can be adjusted depending on the implementation.

Further, Table 2 shows the used language instructions for each task in GoalLadder. The same template is used for RL-VLM-F for controlled comparison. Table 3 shows language instructions used for reward calculation in VLM-RM. Finally, Table 4 shows language instructions used in RoboCLIP.

---

**VLM prompt template in GoalLadder**

Image 1: {IMAGE 1}
Image 2: {IMAGE 2}

The goal {LANGUAGE INSTRUCTION}.

Answer the following questions:
1. What is shown in Image 1?
2. What is shown in Image 2?
3. Is there any difference between Image 1 and Image 2 in terms of achieving the goal?
4. Is the goal better achieved in Image 1 or in Image 2? Explain your reasoning.

If the goal is better achieved in Image 2 than it is in Image 1,
{FORMATTING INSTRUCTIONS}.

---

Table 2: GoalLadder language instructions for each task

| Task | LANGUAGE INSTRUCTION |
|---|---|
| cartpole | "is pole balanced upright" |
| mountaincar | "is car at the peak of the mountain, to the right of the yellow flag" |
| drawer-open-v2 | "of the robotic arm is to open the green drawer" |
| drawer-close-v2 | "of the robotic arm is to close the green drawer" |
| sweep-into-v2 | "of the robotic arm is to sweep the green cube into the hole" |
| window-open-v2 | "of the robotic arm is to open the window" |
| button-press-topdown-v2 | "of the robotic arm is to press the red button into the orange box" |

Table 3: VLM-RM language instructions for each task

| Task | LANGUAGE INSTRUCTION |
|---|---|
| cartpole | "pole vertically upright on top of the cart" |
| mountaincar | "car at the peak of the mountain, to the right of the yellow flag" |
| drawer-open-v2 | "open drawer" |
| drawer-close-v2 | "closed drawer" |
| sweep-into-v2 | "green cube in a hole" |
| window-open-v2 | "open window" |
| button-press-topdown-v2 | "pressed button" |

Table 4: RoboCLIP language instructions for each task

| Task | LANGUAGE INSTRUCTION |
|---|---|
| drawer-open-v2 | "robot opening drawer" |
| drawer-close-v2 | "robot closing drawer" |
| sweep-into-v2 | "robot pushing green cube into a hole" |
| window-open-v2 | "robot opening window" |
| button-press-topdown-v2 | "robot pressing red button" |

# C    Additional ablations

## C.1    ELO rating

To disentangle the contributions of the ELO ranking more thoroughly, we have run a small ablation study of GoalLadder's performance without the ELO system, i.e. the 'greedy' rating system where we always trust the VLM's decision to replace the top-rated goal. Table 5 shows the results on the *Drawer Open* task, confirming the importance of the ELO rating system in achieving good performance.

Table 5: ELO ablation.

| GoalLadder | Final success rate |
|---|---|
| with ELO | $0.97 \pm 0.11$ |
| without ELO | $0.20 \pm 0.35$ |

Qualitatively, we observe the expected suboptimal behaviour in GoalLadder without ELO: significantly better goals are periodically replaced by significantly worse goals as targets (i.e. VLM makes mistakes). This destabilises the training process and results in a rapid performance drop.

## C.2 Buffer size

In our initial experimentation, we have found that the buffer size should be big enough to allow for diversity, but small enough to allow for frequent pairwise comparisons between the candidate goals.

Table 6: Success rates by buffer size.

| Buffer size | Final success rate | Average success rate | Max success rate |
|---|---|---|---|
| 1 (No ELO) | $0.20 \pm 0.35$ | $0.14 \pm 0.23$ | $0.25 \pm 0.45$ |
| 5 | $0.83 \pm 0.13$ | $0.48 \pm 0.13$ | $1.00 \pm 0.00$ |
| 25 | $0.80 \pm 0.11$ | $0.47 \pm 0.14$ | $1.00 \pm 0.00$ |
| 50 | $1.00 \pm 0.00$ | $0.61 \pm 0.11$ | $1.00 \pm 0.00$ |
| 10000 | $0.00 \pm 0.00$ | $0.01 \pm 0.01$ | $0.13 \pm 0.27$ |

To confirm this intuition, we have run a small ablation of GoalLadder with varying buffer sizes. The results shown in Table 6 confirmed our initial hypothesis: (a) smaller buffer sizes hinder performance as the diversity of candidate goals in the buffer suffers; (b) large buffer sizes require an exponentially larger number of pairwise comparisons to arrive at converged ratings, thus also hindering the performance of the model. Overall, we find that GoalLadder is robust to buffer sizes within the reasonable ranges.

## C.3 Visual encoders

We have similarly tested the performance of GoalLadder with two off-the-shelf visual encoders – DINOv2 [38] and CLIP [5] – shown in Table 7. These results indicate that a small VAE trained on environment observations outperforms large, pre-trained encoders. We hypothesise that the VAE allows the model to better distinguish between fine-grained state differences, which are not well captured by the pre-trained models.

Table 7: Success rate by vision encoder.

| Model | Max success rate |
|---|---|
| GoalLadder + VAE | $1.00 \pm 0.00$ |
| GoalLadder + DINOv2 | $0.25 \pm 0.21$ |
| GoalLadder + CLIP | $0.38 \pm 0.17$ |

