# OpenReview forum: "GoalLadder: Incremental Goal Discovery with Vision-Language Models"
_NeurIPS.cc/2025/Conference — NeurIPS 2025 poster_

### Official Review · Reviewer_Y7Jk · 2025-06-28

**Clarity:** 4
**Significance:** 3
**Originality:** 3
**Rating:** 5
**Confidence:** 2

**Summary:**

This paper presents GoalLadder, a RL framework for solving robotics control tasks from a single natural language intruction (without using the dense reward function). The method incrementally discovers goal states using pairwise comparisons from a vision-language model (VLM) and ranks these states using an Elo style rating system. The top-ranked goals are then used to define a reward function based on euclidean distance in a learned latent space. Experimental results show good performance on classical control and and simple manipulation tasks (drawer open, window close etc).

**Questions:**

- How did you robustly verify that the latent space aligns with actual task progress? Did you test whether visually close but semantically different states are rewarded the same? as in doing any edge-case testing.
- What are the compute/runtime implications of GoalLadder vs. preference-based methods? Can the method run efficiently in real-time or on physical robots?

**Ethical Concerns:**

["NO or VERY MINOR ethics concerns only"]

**Limitations:**

- This method only works on tasks where success is clearly apparent from a single image frame.
- Visual similarity is not always equivalent to task completion, this assumption can fail in tasks with subtle state changes.
- This method's robustness is also tied to the accuracy of the VAE learned representation.

**Quality:**

3

**Strengths And Weaknesses:**

## Strengths:
- No handcrafted or dense reward required: As far as I understood, this approach does not require a dense reward, not requires training a reward model
- The writing is clear and the method seems sound.
- Good performance results.

## Weaknesses:
- I'm not fully up-to-date with the SOT in this line of work, but I know some existing baselines in language-conditioned RL such as SayCan [1], Code-as-Policies [2], Language-to-Reward [3]. My general feeling is that these are strongly related work and that should be included as a comparison.
- Text version of RoboCLIP is used even though it also provides a video-level alignment capability.
- This method assues the task completion can be inferred from a single image (it may not work for more complex tasks).
- Reward is defined as euclidean distance in the latent space. This assumes that visual proximity in this space corresponds to task progress and this can be a weak reward proxy in more complex environments.
- No real world experiments.




[1] - https://say-can.github.io/

[2] - https://arxiv.org/pdf/2209.07753

[3] - https://arxiv.org/pdf/2306.08647

---

> ### Author Rebuttal · Authors · 2025-07-30
>
> Dear Reviewer Y7jk,
>
> Thank you for your positive review of our paper!
>
> .
> # TL;DR:
> ---
> - We acknowledge the mentioned works [1, 2, 3]. We clarify that the suggested baselines assume access to control APIs, pre-trained skills, or expert demonstrations. This significantly diverges from our setting where agents learn low-level behaviours directly from VLM feedback in visual environments. We do cite [3] and will cite [1,2] in the final version of the paper.
> - Regarding RoboCLIP, we used the text version to ensure fair comparison, since GoalLadder and its baselines operate *without expert demonstrations* and learn to solve tasks *using only a single language instruction*.
> - Regarding VAE embedding space visualisation, we observed meaningful clustering of the latent space, supported by t-SNE visualisations and a strong negative correlation between latent distances and ground-truth rewards.
> - While both GoalLadder and preference-based methods are similar in runtime inference cost, GoalLadder requires significantly fewer VLM queries, thus making it more scalable and practical for real-world robotics.
>
> We hope our clarifications have addressed your questions. If our response has helped to resolve any of your doubts regarding the paper, we would be very grateful if you would consider increasing your confidence score.
>
> .
>
> # Detailed response
> ---
>
> - **W1: "... some existing baselines in language-conditioned RL such as SayCan [1], Code-as-Policies [2], Language-to-Reward [3]. My general feeling is that these are strongly related work and that should be included as a comparison."**
>
> Thank you for these pointers. While these works deal with language-conditioned RL, their problem settings fundamentally differ from ours. For our baselines, we have selected methods that match the crucial assumptions / problem setting used in GoalLadder; namely, (a) visual environments, (b) access to a language instruction that describes the task at hand, (c) learning without any access to agent's action spaces (i.e. no grounding) or to environment simulation.
>
> Allow us to briefly expand on why we believe it would not be appropriate to use the aforementioned methods as baselines.
>
> In SayCan [1], the agent is assumed to have access to a set of low-level skills, while the LLM is used to select high-level actions (skills) to be executed. In contrast, GoalLadder and its baselines do not assume access to pre-trained behaviours and are concerned with learning low-level behaviours from VLM feedback, instead.
>
> Code-as-Policies [2] does not address the learning of a low-level RL policy and assumes access to control APIs/primitives. It generates code that parametrises pre‑existing perception and control APIs (scripted primitives or motion controllers). In contrast, GoalLadder and its baselines learn behaviours directly in the environment’s intrinsic action space.
>
> Language‑to‑Rewards [3] similarly does not train a low‑level RL policy. Instead, it maps language into a predefined reward library and uses MuJoCo MPC with privileged state / dynamics to optimise actions.
>
> We want to note that we did cite [3] in the Related Work section, pointing out the differences stated above. We would be happy to cite [1] and [2] in the final version of the paper, as well.
>
>
>
> - **W2: "Text version of RoboCLIP is used even though it also provides a video-level alignment capability."**
>
> This choice is justified in the paper (see Experiments: Baselines). To reiterate, in order to ensure a fair comparison (namely, GoalLadder and its baselines must be able to learn using *only a single language instruction*), the text version of RoboCLIP was used.
>
> Using the video version of RoboCLIP would imply access to an expert demonstration, which GoalLadder and its baselines do not have or need by design.
>
>
> - **Q1: "How did you robustly verify that the latent space aligns with actual task progress?"**
>
> This is a great question. In the paper, we implicitly address this question by demonstrating the superior performance of GoalLadder against the baselines, showing that the *resultant rewards are conducive to solving the tasks*.
>
> Nevertheless, to show this more explicitly, we have now (1) made a t-SNE visualisation of the learned latent space while overlaying a colour map of the ground-truth reward and the L2 distance to the top-rated goal; (2) calculated the correlation between the latent L2 distances and the ground-truth reward using subsets of observations from the replay buffer.
>
> For (1) we find that the visualised latent space contains meaningful clustering of the state space. For example, for the `Drawer Open` task, states in which the drawer is open are clustered together and have both high ground-truth reward and low L2 distance to the top-rated goal.
>
> For (2), we find a moderate-to-strong negative Pearson correlation of −0.55 ± 0.007 (r² ≈ 0.30) and a Spearman rank correlation of −0.43 ± 0.01 (p < 1e−16). This shows that the resultant latent distance is a meaningful (though not perfect) proxy for task progress.
>
> We believe both of these results substantiate the fact that GoalLadder produces a useful reward landscape, which is further evidenced by its superior performance against baseline methods.
>
>
> - **Q2: "What are the compute/runtime implications of GoalLadder vs. preference-based methods? Can the method run efficiently in real-time or on physical robots?"**
>
> Thank you for this question.
>
> Both GoalLadder and preference-based methods require the use of a neural network to compute the reward. In the case of GoalLadder -- it is the visual feature extractor and the subsequent calculation of the distance to the top-rated goal. In the case of preference-based methods -- it is the trained reward function. In this sense, there is no substantial difference between the two approaches. The computed rewards are used to train a light-weight SAC policy, which can be deployed on a physical robot.
>
> Where important savings come into the picture is during the *acquisition of the expensive VLM feedback*. As demonstrated, GoalLadder requires substantially fewer VLM queries to solve the tasks. We find this result particularly important if GoalLadder were to be scaled or applied for training physical robots. Preference-based methods would likely require significantly more feedback during the training phase.

---

> ### Comment · Area_Chair_VtYY · 2025-08-06
>
> Please remember to respond to authors' rebuttal as soon as possible.
>
> Thank you!
>
> -AC

---

### Official Review · Reviewer_sgKz · 2025-07-02

**Clarity:** 3
**Significance:** 2
**Originality:** 2
**Rating:** 4
**Confidence:** 3

**Summary:**

This paper introduces GoalLadder, a reinforcement learning (RL) framework that learns from natural language task descriptions using vision-language models (VLMs). Rather than relying on dense reward signals or extensive VLM queries, GoalLadder incrementally builds a ranked buffer of goal candidate states using pairwise VLM comparisons guided by an ELO rating system. The agent is trained to minimize the distance in a learned embedding space to the top-rated goal. This enables sample-efficient, noise-tolerant goal discovery without explicit supervision. The method outperforms prior VLM-based baselines and achieves performance close to or better than oracle reward baselines on standard continuous control and robotic manipulation tasks.

**Questions:**

1. What was the prompt used for the VLM during pairwise comparisons in the discovery phase (i.e., for $\text{VLM}(o^*, o_j, 1)$)?

2. When a new observation $o_j$ is inserted into the goal buffer after outperforming the current best candidate, how is its initial ELO rating determined? i.e., what is $e_j$ if $\text{VLM}(o^*, o_j, 1) = 1$ in the discovery phase?

3. How does the size of the goal buffer affect the performance?

4. The paper uses Gemini 2.0 Flash as the VLM backbone. Have the authors considered how GoalLadder would perform with more capable or recent models, such as GPT-4, Claude, or Gemini 2.5?

5. The use of ELO to refine candidate goal rankings is a core component of GoalLadder, but its added value is not explicitly demonstrated. For instance, have the authors measured the correlation (e.g., Spearman rank correlation) between initial VLM comparisons and final ELO rankings? If the initial discovery decisions are highly predictive of final rankings, this might suggest that the ELO system introduces limited additional robustness.

6. Have the authors observed any failure cases where the ELO system converges to suboptimal goals due to VLM inconsistencies?

7. The method uses a variational autoencoder (VAE) to learn the latent space in which rewards are defined. Have the authors experimented with alternative approaches such as contrastive learning?

8. Since the reward function depends on distances in the learned latent space, it would be helpful to see an analysis of whether this space meaningfully reflects task progress. Have the authors performed any sanity checks such as t-SNE or PCA visualizations of the latent representations?

**Ethical Concerns:**

["NO or VERY MINOR ethics concerns only"]

**Final Justification:**

NA

**Limitations:**

Yes.

**Paper Formatting Concerns:**

None.

**Quality:**

2

**Strengths And Weaknesses:**

**Strengths**

* Achieves strong empirical results: ~95% average success rate, outperforming all prior VLM-based baselines by a wide margin.

* Significantly more sample-efficient in terms of VLM queries (∼4500 vs. ∼15000 for PEBBLE).

* Clear, well-organized presentation with helpful diagrams and detailed algorithm description.

* Leverages a learned embedding space to define rewards, enabling generalization to unseen states without requiring dense supervision.


**Weaknesses**

* No appendix is given in the paper. Should be fixed.

* Embedding space is learned via VAE, but no comparison to alternative methods (e.g., contrastive learning).

* Uses only Gemini 2.0 Flash as the VLM backbone; lacks evaluation on robustness across different models.

* No analysis of robustness under adversarial or inconsistent VLM feedback.

* The current evaluation is limited to five Metaworld tasks. In contrast, RL-VLM-F evaluated on a wider set, including Soccer, Cloth Fold, Straighten Rope, and Pass Water.

---

> ### Author Rebuttal · Authors · 2025-07-30
>
> Dear Reviewer sgKz,
>
> Thank you for your review!
>
> .
>
> # TL;DR:
> ---
> - We clarify that the appendix was included in the supplementary zip and already addresses several of your questions, including full prompt details (Q1) and ELO initialisation (Q2).
> - We've now run ablations on goal buffer size (Q3). GoalLadder is robust to reasonably sized buffer sizes (5-500). Small (<5) and large (>500) buffer sizes deteriorate performance in line with the intuition. **See Table 1**.
> - We've now run ablations to confirm that ELO ranking significantly improves robustness  over naive ranking, thus disentangling the contribution of the ELO rating system (Q5). **See Table 2.**
> - We have observed no notable failure cases from VLM inconsistencies *after* adding the ELO rating system (Q6).
> - While we used Gemini 2.0 Flash due to financial constraints, GoalLadder is designed to  perform better with better VLMs (Q4). In our opinion, using the less performative VLM adds to the significance of our results.
> - Regarding VAE embedding space visualisation (Q7–8), we observed meaningful clustering of the latent space, supported by t-SNE visualisations and a strong negative correlation between latent distances and ground-truth rewards. We welcome future work attempting to use contrastive learning to improve the embedding space further.
>
> We hope our clarifications and new experiments have addressed your questions. In light of these updates, we would be very grateful if you would consider updating your evaluation. We believe the revised version merits a higher score, especially considering the fact that the initial review did not take into account the appendix provided. We think it may have affected your scores under "*quality*" and "*clarity*", given that the other reviewers provided scores of at least 3 for both.
>
> .
>
> # New results
> ---
> Tested on the `Drawer Open` task with 3 random seeds.
>
> ---
>
> >**Table 1**: Buffer size.
>
> | Buffer size | Final success rate | Average success rate | Max success rate |
> | :---------: | :----------------: | :------------------: | :--------------: |
> | 1 (No ELO)  |    0.20 ± 0.35     |     0.14 ± 0.23      |   0.25 ± 0.45    |
> |      5      |    0.83 ± 0.13     |     0.48 ± 0.13      |   1.00 ± 0.00    |
> |     25      |    0.80 ± 0.11     |     0.47 ± 0.14      |   1.00 ± 0.00    |
> |     50      |    1.00 ± 0.00     |     0.61 ± 0.11      |   1.00 ± 0.00    |
> |     500     |    0.75 ± 0.24     |     0.56 ± 0.06      |   1.00 ± 0.00    |
> |    10000    |    0.00 ± 0.00     |     0.01 ± 0.01      |   0.13 ± 0.27    |
>
> ---
>
> >**Table 2**: ELO rating system.
>
> | GoalLadder  | Final success rate |
> | :---------: | :----------------: |
> |  with ELO   |    0.97 ± 0.11     |
> | without ELO |    0.20 ± 0.35     |
>
>
> .
>
> # Detailed response
> ---
>
> - **"No appendix is given in the paper. Should be fixed."**
>
> We apologise for the confusion -- we did in fact provide the appendix; however, it was attached separately in the supplementary materials zip file. Indeed, several of the questions you raised in the review are answered in the appendix. It includes: implementational details, computational resources, prompts.
>
>
> - **Q1: "What was the prompt used for the VLM during pairwise comparisons in the discovery phase?"**
>
> The full prompt can be found in the appendix. In short, we ask the model to compare two images with respect to a specified text instruction.
>
>
> - **Q2: "How is its initial ELO rating determined?"**
>
> This is similarly clarified in the appendix. In short, the initial rating of a newly-added candidate goal is computed to be the average of ELO ratings of other goals in the buffer (i.e., the goal is inserted in the middle of the buffer). This ensures that newly-discovered goals are not immediately considered to be superior to the top-rated goal.
>
>
> - **Q3: "How does the size of the goal buffer affect the performance?"**
>
> This is a great question. In our initial experimentation, we have found that the buffer size should be big enough to allow for diversity, but small enough to allow for frequent pairwise comparisons between the candidate goals.
>
> Following your suggestion, we have now run an ablation study with varying buffer sizes. The results confirmed our initial hypothesis: (a) smaller buffer sizes hinder performance as the diversity of candidate goals in the buffer suffers; (b) large buffer sizes require an exponentially larger number of pairwise comparisons to arrive at converged ratings, thus also hindering the performance of the model -- **see Table 1.**
>
>
> - **Q4: "Have the authors considered how GoalLadder would perform with more capable or recent models, such as GPT-4, Claude, or Gemini 2.5?"**
>
> It is unfortunate that experiments with commercial VLMs require substantial financial resources. As such, we were unable to evaluate the performance of GoalLadder using the more expensive models. We hope for your understanding in this matter.
>
> However, we would like to stress two points here: (1) the existing VLMs still struggle with spatial understanding (e.g., [1]). In our small-scale experimentations, we have found that they fall prey to the same problems as Gemini 2.0 Flash. (2) Perhaps more importantly, GoalLadder is designed to perform better with better VLMs. This is because more accurate VLM labels would result in the faster convergence of goal ratings to their 'true' ratings. GoalLadder will retain its capability to solve the tasks studied in the paper, while also maintaining its superior sample efficiency against the competitor methods. Even in its current form, GoalLadder produces *better sample efficiency than PEBBLE*, which uses ground-truth preference labels -- *effectively amounting to a VLM producing labels with 100% accuracy*. We believe this result substantiates the above claim.
>
> [1] SpatialVLM, Chen et al., 2024
>
>
> - **Q5: "The use of ELO to refine candidate goal rankings is a core component of GoalLadder, but its added value is not explicitly demonstrated."**
>
> This is a great question -- we appreciate that you brought it up. In the paper, we address this point by having a controlled comparison with RL-VLM-F, in which inconsistent or noisy feedback is not explicitly dealt with. Since it failed to solve most of the tasks, we believe it is a reasonable indicator of said issue.
>
> Nevertheless, to address your question more thoroughly, we have run a small ablation study of GoalLadder's performance *without* the ELO system, i.e. the 'greedy' rating system where we always trust the VLM's decision to replace the top-rated goal. As another reviewer pointed out, this would effectively showcase the contribution of the ELO system to our model.
>
> Having run this ablation on the Drawer Open task, we can confirm that the ELO rating system is a crucial component of GoalLadder -- **see Table 2.**
>
> Qualitatively, we observe the expected suboptimal behaviour: significantly better goals are periodically replaced by significantly worse goals as targets (i.e. VLM makes mistakes). This destabilises the training process and results in a rapid performance drop.
>
>
> - **Q6: "Have the authors observed any failure cases where the ELO system converges to suboptimal goals due to VLM inconsistencies?"**
>
> In our initial experimentations, particularly with RL-VLM-F, we have observed that VLMs can make frequent and often consistent mistakes in some pairwise comparisons. We therefore noted that trusting VLMs completely may result in such 'suboptimal' convergence points. Based on the experiments we ran since the incorporation of ELO, we have not observed notable failure cases, as diverse pairwise comparisons allow VLMs to get out of these  suboptimal loops, when and if they appeared. This is further evidenced by the ablation study we ran in regard to your Q5.
>
>
> - **Q7: "Have the authors experimented with alternative approaches such as contrastive learning?" *AND* "Embedding space is learned via VAE, but no comparison to alternative methods (e.g., contrastive learning)."**
>
> In the paper, we acknowledge that more sophisticated methods for representation learning, such as contrastive learning, would be interesting to investigate in future work.
>
> However, the results show clear evidence that a simple visual feature extractor suffices to produce state-of-the-art results in this area. Self-supervised learning techniques may be used to *extend* our method by considering temporal ordering or other learning objectives to improve the landscape of the latent space. Still, we believe that the use of a VAE does not diminish the significance of our results.
>
>
> - **Q8: "It would be helpful to see an analysis of whether this space meaningfully reflects task progress."**
>
> This is a useful suggestion. In the paper, we implicitly address this question by demonstrating the superior performance of GoalLadder against the baselines, showing that the *resultant rewards are conducive to solving the tasks*.
>
> Nevertheless, we have (1) made a t-SNE visualisation of the learned latent space while overlaying a colour map of the ground-truth reward and the L2 distance to the top-rated goal; (2) calculated the correlation between the latent L2 distances and the ground-truth reward using subsets of observations from the replay buffer.
>
> For (1) we find that the visualised latent space contains meaningful clustering of the state space. For example, for the `Drawer Open` task, states in which the drawer is open are clustered together and have both high ground-truth reward and low L2 distance to the top-rated goal.
>
> For (2), we find a moderate-to-strong negative Pearson correlation of −0.55 ± 0.007 (r² ≈ 0.30) and a Spearman rank correlation of −0.43 ± 0.01 (p < 1e−16). This shows that the resultant latent distance is a meaningful (though not perfect) proxy for task progress.
>
> We believe both of these results substantiate the fact that GoalLadder produces a useful reward landscape, which is further evidenced by its superior performance against baseline methods.

---

> > ### Comment · Reviewer_sgKz · 2025-08-05
> > **nan**
> >
> > my major worries have been addressed, especially with the presence of the appendix and the new experiments.

---

> > > ### Author Response · Authors · 2025-08-05
> > >
> > > Dear Reviewer sgKz,
> > >
> > > We appreciate you getting back to us and we are pleased that we were able to address your questions in our rebuttal.

---

### Official Review · Reviewer_zeqD · 2025-07-03

**Clarity:** 3
**Significance:** 3
**Originality:** 3
**Rating:** 4
**Confidence:** 4

**Summary:**

1. This paper introduces GoalLadder, a novel method for training RL agents in visual environments from a single natural language instruction. The core idea is to leverage a VLM to incrementally discover and rank goal states that represent progress towards the final task.
2. This approach avoids the need for a large volume of VLM queries and the direct generation of a reward function, making it more query-efficient and robust.
3. The authors demonstrate through experiments on classic control and robotic manipulation tasks that GoalLadder significantly outperforms existing methods and nearly matches, or surpasses, the performance of an oracle agent with access to the ground-truth reward.

**Questions:**

1. The paper's strongest component seems to be its robustness to noisy VLM feedback, which the authors attribute to the ELO rating system. Could you provide an ablation study or further analysis comparing the ELO system to a simpler ranking mechanism (e.g., a "greedy" approach that always replaces the current best goal with any new goal favored by the VLM)? This would help quantify the direct impact of the ELO system.
2. The result where GoalLadder surpasses the oracle agent on the "Drawer Open" task is fascinating. The paper suggests this is due to the difficulty of manual reward engineering. Could the authors provide more insight into this? For example, what does the discovered goal image ($o^*$) look like compared to the state that would yield the maximum ground-truth reward? Does GoalLadder discover a reward function that provides a better-shaped gradient for the policy to learn?
3. The reward function is defined as the negative Euclidean distance in the latent space of a VAE trained on collected observations. What is the intuition behind why this is a reasonable approach? This implicitly assumes that Euclidean distance in this learned space is a meaningful proxy for task progress. Have you considered or experimented with alternative distance metrics or using a different unsupervised representation learning objective (e.g., contrastive learning) instead of a VAE?
4. How did you choose the key hyperparameters like the goal buffer size, query frequency (K), and reward update frequency (L)? A brief discussion on the sensitivity to these parameters would be valuable for others seeking to reproduce or build upon your work. For instance, is there a trade-off between VLM query cost and performance based on the choice of K and M?
5. Could you discuss the potential for extending this method to more complex tasks, for instance, tasks where it is notoriously difficult to engineer a reward function? Furthermore, could you add an analysis and discussion of the method's failure cases? Clarifying the boundaries of this method will help the community understand the problem's difficulties and the paper's contribution.

**Ethical Concerns:**

["NO or VERY MINOR ethics concerns only"]

**Final Justification:**

I sincerely thank the authors for their detailed response and the effort made during the rebuttal period. All of my concerns have now been addressed. I have also read the discussions between the authors and other reviewers.

While I believe this work makes a valuable contribution to better utilizing pre-trained knowledge for reward modeling, I have some reservations about its broader impact. The method's applicability does not seem to extend beyond the scope of prior work (e.g., RoboCLIP), and its underlying mechanism remains unclear. As the authors themselves note, the use of the L2 norm is an empirical finding rather than a theoretically motivated choice. For these reasons, the work provides a useful result but lacks deeper, more generalizable principles.

Therefore, I recognize the value of this specific work. I hope the paper is accepted by the community. I also encourage the authors to explore more principled methods for leveraging pre-trained knowledge in future iterations. I will maintain my positive evaluation.

**Limitations:**

yes

**Quality:**

3

**Strengths And Weaknesses:**

**Strengths:**

1. The primary contribution is a novel approach to learn behaviors from a natural language instruction by leveraging VLM. The use of an ELO rating system to filter out noise from VLM judgments is a significant and original idea in this context. This addresses a well-known challenge of VLM-based feedback systems. Furthermore, the method decouples VLM feedback from the dense reward signal by using the VLM for goal-ranking and a separately trained autoencoder for defining a distance-based reward metric.
2. The paper demonstrates exceptional performance. GoalLadder achieves an average success rate of \~95% across all tasks, dramatically outperforming the next best baseline (\~45%).
3. Its high sample efficiency with respect to expensive VLM queries makes the approach more practical and scalable than many existing alternatives.
4. The paper is well-written, clearly structured, and easy to follow. The method is explained logically, and the inclusion of Figure 1 and Algorithm 1 provides an excellent high-level overview of the system's components and data flow. The motivation and problem statement are well-established in the introduction.

**Weaknesses:**

1. The paper is missing ablation studies that would help disentangle the contributions of its key components. For example:
   - How critical is the ELO rating system? How does it compare to a simpler approach, such as always trusting the VLM's judgment or averaging scores? An ablation here would strengthen the claim about robustness to noise.
   - How important is learning a visual representation via a VAE? How would the system perform if the reward was calculated using distances in the VLM's own embedding space (e.g., CLIP's), or another off-the-shelf visual encoder?
2. The method introduces several new hyperparameters, including the goal buffer size (∣Bg∣=10), the VLM query frequency (K), the number of queries (M), and the reward update schedule (L). The paper does not include a sensitivity analysis, making it unclear how difficult the method might be to tune for new environments.
3. While the experiments are strong, they are confined to simulation environments with relatively clear visual progress. The authors acknowledge that the method assumes static goals identifiable from a single image. It is unclear how well the VAE-based visual similarity metric would work in tasks where semantic understanding is more critical than visual feature matching (e.g., "put the red block on the blue block" if there are other visually similar blocks).

---

> ### Author Rebuttal · Authors · 2025-07-30
>
> Dear Reviewer zeqD,
>
> Thank you for your positive review of our paper. In what follows, we clarify a few points and present additional ablation results you recommended to include.
>
> .
>
> # TL;DR:
> ---
> - We ran ablations to confirm that ELO ranking significantly improves robustness over naive ranking, thus disentangling the contribution of the ELO rating system (W1 / Q1). **Table 2**.
> - We ran ablations showing that GoalLadder + VAE outperforms off-the-shelf encoders  like CLIP/DINOv2, which fail to capture fine visual differences and produce embedding spaces with weaker task alignment (W1). **Table 3.**
> - We explain why GoalLadder can sometimes outperform the oracle agent (Q2): ground-truth reward functions include implicit constraints on *how* tasks are expected to be solved; GoalLadder does not constrain the space of solutions.
> - On reward shaping (Q3), we clarify on our initial experimentation with the different reward metrics (L1, L2, cosine similarity). We also agree that contrastive learning is a promising future direction.
> - On hyperparameters (Q4), we present a buffer size ablation showing clear performance trade-offs. Other hyperparameters were selected to ensure a controlled comparison with the baselines. **Table 1.**
> - Finally, we reiterate our method’s limitations (Q5) and outline future work, such as extending to dynamic goals and improving embeddings via self-supervision.
>
> We hope our clarifications and new experiments have addressed your questions. In light of these updates, we would be grateful if you would consider increasing your evaluation of our paper.
>
> .
>
> # New results
> ---
> Ablations tested on the `Drawer Open` task with 3 random seeds.
>
> ---
>
> >**Table 1**: Buffer size.
>
> | Buffer size | Final success rate | Average success rate | Max success rate |
> | :---------: | :----------------: | :------------------: | :--------------: |
> | 1 (No ELO)  |    0.20 ± 0.35     |     0.14 ± 0.23      |       0.25 ± 0.45       |
> |      5      |    0.83 ± 0.13     |     0.48 ± 0.13      |       1.00 ± 0.00     |
> |     25      |    0.80 ± 0.11     |     0.47 ± 0.14      |       1.00 ± 0.00      |
> |     50      |    1.00 ± 0.00     |     0.61 ± 0.11      |       1.00 ± 0.00      |
> |     500     |    0.75 ± 0.24     |     0.56 ± 0.06      |       1.00 ± 0.00      |
> |    10000    |    0.00 ± 0.00     |     0.01 ± 0.01      |       0.13 ± 0.27      |
>
> ---
>
> >
> >**Table 2**: ELO rating system.
>
> | GoalLadder  | Final success rate |
> | :---------: | :----------------: |
> |  with ELO   |    0.97 ± 0.11     |
> | without ELO |    0.20 ± 0.35     |
>
> ---
>
> >**Table 3**: Visual encoders.
>
> |        Model        | Max success rate (%) |
> | :-----------------: | :------------------: |
> |  GoalLadder + VAE   |     1.00 ± 0.00      |
> | GoalLadder + DINOv2 |     0.25 ± 0.21      |
> |  GoalLadder + CLIP  |     0.38 ± 0.17      |
>
> .
>
> # Detailed response
> ---
>
> - **W1 / Q1: "How critical is the ELO rating system? How does it compare to a simpler approach, such as always trusting the VLM's judgment or averaging scores?"**
>
> This is a great question. In the paper, we address this point by having a controlled comparison with RL-VLM-F. As mentioned, in RL-VLM-F, inconsistent or noisy feedback is not explicitly dealt with. Since the results indicate that RL-VLM-F fails to solve most of the tasks, we believe it is a reasonable indicator of said issue.
>
> Nevertheless, as you suggested, we have run a small ablation of GoalLadder's performance _without_ the ELO system, i.e. the 'greedy' rating system where the VLM's decision to replace the top-rated goal is always trusted. Ablation results on the `Drawer Open` task confirm our hypothesis -- **Table 2**.
>
> Qualitatively, we observe the expected suboptimal behaviour: better goals are periodically replaced with worse goals as targets (i.e. VLM makes mistakes). This destabilises the training process and results in a rapid performance drop.
>
>
> - **W1: "How important is learning a visual representation via a VAE?"**
>
> Thank you for this question. In GoalLadder, we attempted to use one of the simplest examples of a visual feature extractor fine-tuned to the environment -- in the form of a VAE. Still, as we mention in the discussion section, we suggest future work to investigate self-supervised learning techniques to potentially improve the embedding space further.
>
> Following your suggestion, we have tested the performance of GoalLadder with two visual encoders: DINOv2 (Oquab et al., 2023) and CLIP (Radford et al., 2021) on the `Drawer Open` task -- **Table 3**. Further, we visualised the embedding space produced by DINOv2 / CLIP using t-SNE and overlayed the ground-truth rewards as a colour map, and compared it to the embedding space produced by our trained VAE model.
>
> What we found is in line with the initial hypothesis. Firstly, GoalLadder + DINOv2 / CLIP backbone performs worse on the tested task. The plotted visual embedding space of DINOv2 and CLIP with ground-truth reward colour mapping indicates poor clustering: (a) there exist a large number of visual states that produce a low Euclidean distance (high GoalLadder reward) to the top-rated goal, and (b) high ground-truth reward states are significantly more spread out compared to the embedding space produced by the VAE.
>
> Nevertheless, as you point out, we think that the use of off-the-shelf visual encoders may be useful in environments where fine visual differences are not crucial *and/or* where semantic differences are of particular importance. We leave this investigation for future work.
>
>
> - **Q2: "The result where GoalLadder surpasses the oracle agent on the "Drawer Open" task is fascinating. Could the authors provide more insight into this?"**
>
> Thank you for acknowledging this, we also think this result is particularly interesting.
>
> What we believe is crucial is *how* the agent gets to the success state. Manually-designed Metaworld reward functions contain some constraints on how the task is expected to be solved. For instance, in the `Drawer Open` task, the agent is rewarded for grasping the handle; however, in many cases, the agent is able to open the drawer using the *side* of the handle. As such, with GoalLadder, the agent has more options on how to solve the task. Similarly, in the `Sweep Into` task, the reward function rewards the grasping of the cube; however, this is not necessary to solve the task, since pushing the cube can be as effective.
>
>
> - **Q3: "The reward function is defined as the negative Euclidean distance in the latent space of a VAE trained on collected observations. What is the intuition behind why this is a reasonable approach?"**
>
> Thank you for this point. We have indeed experimented with (a) L1 distance, (b) L2 distance, and (c) cosine similarity. Upon initial experimentation L2 distance worked best -- hence our choice. We were unable to run large-scale ablation experiments due to financial constraints in using commercial VLMs. With regard to alternative learning objectives, we agree that it is a promising direction of improving GoalLadder even further. At this point, we leave this direction for future work.
>
>
> - **Q4: "How did you choose the key hyperparameters like the goal buffer size, query frequency (K), and reward update frequency (L)?"**
>
> In order to disentangle the contribution of the novel components in GoalLadder against its best competitor model (RL-VLM-F), we wanted to ensure a controlled comparison. For this, we used the same hyperparameters and feedback scheduling format, where possible. As such, we fixed the reward function updates to L = 5,000 (used in PEBBLE (Lee et al., 2021) and RL-VLM-F). For the same reason, we used the standard RLHF feedback scheduling format by fixing K (query frequency) and M (feedback size) over the course of training. This allowed for a more substantiated empirical comparison of the two models.
>
> The feedback rate was chosen in line with the rate of experience accumulation (i.e. episode length, which was 500 for Metaworld), making K = 500. Given the design of the system, we expect it to work better with higher feedback rates; however, as demonstrated, the current feedback rate is: (a) already significantly lower than those of the baseline models; (b) leads to a clearly superior performance against the baselines.
>
> With regard to the buffer size, we have now run an ablation study with varying buffer sizes. Please see the following results for the `Drawer Open` task -- **Table 1.** These results confirmed our initial hypothesis: (a) smaller buffer sizes hinder performance as the diversity of candidate goals suffers; (b) larger buffer sizes require an exponentially larger number of pairwise comparisons for ratings to converge, thus also hindering the performance of the model.
>
>
> - **Q5: "Could you discuss the potential for extending this method to more complex tasks? Furthermore, could you add an analysis and discussion of the method's failure cases?"**
>
> The limitations of our method are discussed in Section 5.1. To reiterate: (a) we assume that the progress or success of a task can be identified from a single image, limiting the application of our method to environments with static goals, and (b) GoalLadder relies on visual feature similarity for reward definition. In some settings, visual similarity between observations could be a limiting proxy for the underlying state similarity or task progress. Here, more advanced visual embedding techniques could be used instead.
>
> Possible extensions are mentioned in Section 5.3. In short, we think that there are two major directions of improvement: (a) video-based evaluation of task progress, where a VLM could be used to point out improving sequences of actions; and (b) improving the visual embedding space with self-supervised learning techniques. The former would allow GoalLadder to work with dynamic goals, while the latter could improve the learning signal from the distance minimisation objective.

---

> ### Comment · Reviewer_zeqD · 2025-08-04
> **Thanks for Detailed Response**
>
> I sincerely thank the authors for their detailed response and the effort made during the rebuttal period. All of my concerns have now been addressed. I have also read the discussions between the authors and other reviewers.
>
> While I believe this work makes a valuable contribution to better utilizing pre-trained knowledge for reward modeling, I have some reservations about its broader impact. The method's applicability does not seem to extend beyond the scope of prior work (e.g., RoboCLIP), and its underlying mechanism remains unclear. As the authors themselves note, the use of the L2 norm is an empirical finding rather than a theoretically motivated choice. For these reasons, the work provides a useful result but lacks deeper, more generalizable principles.
>
> Therefore, I recognize the value of this specific work. I hope the paper is accepted by the community. I also encourage the authors to explore more principled methods for leveraging pre-trained knowledge in future iterations. I will maintain my positive evaluation.

---

> > ### Author Response · Authors · 2025-08-05
> >
> > Dear Reviewer zeqD,
> >
> > Thank you for your response and for reiterating your positive evaluation of our paper. We are happy that we were able to address your questions in our rebuttal.

---

### Note · Authors · 2025-08-13

We thank the reviewers for the constructive discussion. Across all three reviews, the core strengths of our work were consistently acknowledged:
- A novel and robust approach of training RL agents from natural language instructions;
- Strong empirical performance, achieving ~95% success rate and outperforming prior VLM-based methods by a wide margin;
- Significant improvement in VLM query efficiency over prior work.

During the rebuttal, we addressed all raised concerns with new experiments and clarifications. **All** **reviewers** indicated that their major concerns were resolved. Reviewers zeqD and Y7Jk explicitly maintained a clear acceptance recommendation.

As such, we respectfully submit that our paper merits acceptance. We thank the reviewers and the AC for their work!

---

### Decision · Program_Chairs · 2025-09-17

**Decision:**

Accept (poster)

**Comment:**

Authors present a method that uses a VLM to rank subgoals for completing a task using an ELO rating system to 1) reduce the number of queries made to the VLM and 2) avoid learning a parameterized reward function. They also use a pretrained embedding space as a dense reward to gauge progress towards these subgoals. There are strong empirical results that significantly outperform similar methods and significantly reduces the number of required queries to the VLM.

Strengths: Strong empirical performance and improved efficiency in terms of the number of needed queries of the VLM compared to the next best prior method. The paper is clearly written, and the high-level method well laid out.

Weaknesses: There is a lack of ablations testing each component of the method, i.e. the representation used as dense reward for reaching subgoals, the importance of the ELO rating system, and there was a lack of sensitivity analysis to the many new hyperparameters introduced as part of the method.

Most important reasons: There is significant improvement over prior work on multiple fronts (computational needs and overall performance) and one of the main contributions seems to be the use of the ELO rating system, which is an interesting insight.

Reviewer discussion: The weaknesses that were brought up in the reviews were broadly addressed in the rebuttal phase. The authors provided extensive additional ablations and hyperparameter sweeps to demonstrate how their method is affected by various changes. There were concerns brought up about assumptions about the environment that are necessary for this method to work, i.e. obvious visual progress from the observation space and missing environments that were present in prior work. The prior point was part of the authors' discussion about limitations which I commend, but the latter point was not addressed.